# Digital Pseudo-Identification in the Post-Truth Era: Exploring Logical Fallacies in the Mainstream Media Coverage of the COVID-19 Vaccines

Ekaterina Veselinovna Teneva 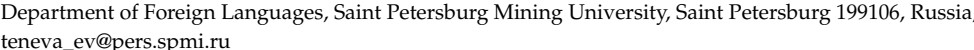

Department of Foreign Languages, Saint Petersburg Mining University, Saint Petersburg 199106, Russia; teneva_ev@pers.spmi.ru

**Abstract:** Because of China's new wave of COVID-19 in May 2023, the issue of tackling COVID-19 misinformation remains relevant. Based on Lippmann's theory of public opinion and agenda setting theory, this article aims to examine the concept of digital pseudo-identification as a type of logical fallacy that refers to supporting journalists' opinions with 'false' arguments that lack factual evidence. To do so, the study applied computer-aided content analysis, as well as rhetorical and critical discourse analyses, to examine 400 articles related to four COVID-19 vaccines ('Oxford-AstraZeneca', 'Pfizer-BioNTech', 'Sputnik V' and 'Sinovac') published on the online versions of two major British and American mainstream media sources between August 2020 and December 2021. The results of the study show that journalists of the 'The New York Times' and 'The Guardian' used similar logical fallacies, including the opinions of pseudo-authorities and references to pseudo-statistics and stereotypes, which contributed to creating distorted representations of the COVID-19 vaccines and propagating online misinformation. The study also reveals political bias in both of the mainstream media sources, with relatively more positive coverage of the European vaccines than non-European vaccines. The findings have important implications for journalism and open up perspectives for further research on the concept of digital pseudo-identification in the humanities and social sciences.

**Keywords:** COVID-19; logical fallacies; digital pseudo-identification; pseudo-authorities; pseudo-statistics; stereotypes; Walter Lippmann; agenda setting; online misinformation; mainstream media

## 1. Introduction

In recent years, global issues such as fighting the COVID-19 pandemic and tackling the sustainable development goals have become some of the major challenges of our times (Pan and Zhang 2020; Yurak et al. 2020). Nevertheless, it is the spread of online misinformation, disinformation and malinformation about COVID-19 that has caused a shift in the form of communication processes and poses a serious threat to modern media and their role in the democratic system (Zarocostas 2020; Muhammed and Mathew 2022; Casero-Ripollés et al. 2023). Although all these concepts generally refer to the proliferation of false information, prior studies revealed discrepancies in their definitions. For instance, Floridi (2013) defined disinformation as misinformation that is purposefully conveyed to mislead the receiver into believing that it is information. Brennen et al. (2020) considered malinformation as 'reconfigured true information', whereas Aïmeur et al. (2023) viewed 'reconfigured' information and 'totally fabricated' information as misinformation. Overall, most studies base the taxonomy of these concepts on such criteria as truth and intentionality. Thus, Wu et al. (2019) distinguished disinformation, which is information that is fake or misleading and spreads intentionally, from misinformation, which is an umbrella term for all false information that is spread unintentionally on social media. Nevertheless, some researchers claim that it is not sufficient for researchers to define misinformation as 'false information' (Baines and Elliott 2020), since 'false information', or 'pseudo-information', is a counterconcept to information and, therefore, misinformation should not be regarded

as information at all (Floridi 2013). A study by Kim and de Zúñiga (2021) suggests that pseudo-information is not contrary to information but is 'an umbrella term for all types of false and inaccurate information (including misinformation and disinformation) that has harmful consequences or social externalities that affect information subscribers. Following this approach, the present study views misinformation as a type of pseudo-information that is spread unintentionally in the media.

The spread of COVID-19 misinformation has accelerated because of several factors, including the rise of citizen journalism (Obama 2023) and the pace of the digitalization of both public and private life (Bylieva et al. 2022; Jaumotte et al. 2023). On the one hand, the strengthening of public discourse, which provided the public more power to express their attitudes and personal opinions, has resulted in excessive public reliance on digital and social media and public misperceptions about COVID-19 (Lee et al. 2023). On the other hand, digitalization has led to various media-related changes, which are commonly referred to as the process of 'deep mediatization' (Hepp 2019). It is a new stage of mediatization in which digital media are embedded into meaningful units of everyday practice so that they are no longer regarded as an 'independent' and discrete social domain (Hepp 2022a). Furthermore, the 'total visualization' of modern media content (Dorofeev and Tomaščíková 2021) has also contributed to misleading audiences about COVID-19, since it has misunderstood some of the most common COVID-19 visualizations (Adkins 2023). Thus, the entanglement of our social world with media technologies has led to a rethinking of the influence of mediated communication on constructing reality (Hepp 2022b) and contributed to the information crisis.

Most recent studies on the information crisis, or the 'post-truth' crisis, where people are more likely to accept an argument based on their emotions and beliefs rather than one based on facts (González-Méijome 2017), have dealt with analyzing the content and spread of fake news. According to Hepp (Hepp cited by Kumar Putta and Anderson 2021), what we know about the coronavirus is communicated to us through the media. Chavda et al. (2022) revealed that during the COVID-19 crisis, people started to believe false news, took home remedies and believed fraudulent health claims on social media. Muhammed and Mathew (2022) also found that misinformation stems from a lack of information from official sources such that people tend to fill this gap with 'improvised news'. Other researchers have gone further and examined the effects of fake news exposure on citizens' behavior (Ognyanova et al. 2020; Casero-Ripollés et al. 2023), revealing that the information crisis is largely due to public distrust towards media (Van Scoy et al. 2021). For instance, Ognyanova et al. (2020) found that fake news exposure was associated with a decline in mainstream media trust among respondents. Likewise, the findings of Casero-Ripollés et al. (2023) showed that disinformation generates an increase in mistrust towards both the media and politicians, which questions the credibility of these two traditional actors in the public sphere. Nevertheless, this has a lower impact on changing the opinion of citizens and their voting decisions, which means that citizens are either becoming used to living in post-truth circumstances in which truth is at risk, or they may be unaware that they are consuming fake news that shapes their attitudes. Thus, the rapid spread of fake news and the confusion concerning the opinions, beliefs and facts about COVID-19 have prompted researchers to readdress the concepts of science and pseudo-science.

According to a study by González-Méijome (2017), science relies on evidence, which is defined as the available body of facts or information, properly collected and analyzed. These facts indicate whether a belief or proposition is true or valid. On the contrary, pseudo-science relies on beliefs, exaggerated or unprovable claims, confirmation bias, lack of openness to evaluation by other experts, and absence of systematic practices when developing theories. Following this approach, the present study suggests that misinformation on COVID-19 in news reporting refers to pseudo-scientific information that is used by journalists unintentionally and lacks scientific and factual evidence. Although the majority of recent studies on COVID-19 have focused either on purely linguistic (Goncharova et al. 2022; Teneva and Bykov 2023) or political features of the COVID-19 discourse (Hart et al.

2020; Van Scoy et al. 2021) and detected COVID-19 misinformation mainly on social media (Wu et al. 2019; Ferrara et al. 2020), less attention has been paid to the issues of tackling COVID-19 misinformation in the mainstream media, which remain under-researched in the context of COVID-19. In contrast to social media, they are regarded as more reliable and have traditional gatekeepers, who crosscheck information sources before publication (Muhammed and Mathew 2022). Therefore, this study aimed to contribute to filling this gap by studying the specific logical fallacies that contribute to propagating online misinformation concerning COVID-19 vaccines in mainstream media from rhetorical, linguistic and journalistic perspectives.

## 2. Theoretical Context

Since the study aimed to detect 'false' information in the COVID-19 vaccine coverage, it was essential to understand the nature of 'false' argumentation and its significant role in our interpretation of the world.

### 2.1. An Overview of Fallacy Studies

Studies of 'false' argumentation or 'logical fallacies', which can simply be defined as defective arguments containing logical errors, date back to Aristotle, who first defined them as hidden arguments which give the illusion to an argument being sound (Hansen 2020). For Aristotle, the art of persuasion is a combination of three main components: appeals to reason ('logos'), emotions ('pathos') and credibility ('ethos'). To convince the listeners, a good speaker should craft his (her) message using not only facts but also emotional appeals, which Aristotle considers to be a means of persuasion (Aristotle cited in Coelho and Huppes-Cluysenaer 2018). Thus, Aristotle highlights the importance of an argument's emotional aspect and overcomes the dualism of rationality and emotionality in argumentation. This approach to the role of emotions in argumentation is reflected in modern media studies, which consider emotions as a means of both persuasion and manipulation that journalists deploy in news production (Glück 2018). Thus, these fundamental appeals to 'logos', 'pathos' and 'ethos' can be viewed not only as a way to persuade the audience but also as a way to manipulate public opinion, depending on the author's intentions. Emotional fallacies play unfairly on emotional appeals, such as those we see in antivaccination campaigns that prey on parent's fear of vaccine-induced damage to their child, despite overwhelming evidence of vaccine safety (Kuchel and Rowland 2023). Ethical fallacies overplay the authority, credibility or character of the messenger, whereas logical fallacies rely on those facts and evidence which are favorable to the author's arguments. The latter finding is significant to our study, which views logical fallacies as a means of manipulating public opinion.

Francis Bacon also contributed to studies on fallacies. He identified the four 'idols of human mind' that prevent people from attaining a true understanding of things, including 'tribe' (our human nature, which distorts our view of the world), 'cave' (our experience, which affects how we interpret the world), 'marketplace' (language as the source of our mistaken ideas) and 'theatre' (false philosophies that rule men's minds) (Hansen 2020). Based on Aristotle's and Bacon's classifications of fallacies, The Port-Royal Logic (cited by Hansen 2020) considered fallacies that are associated with scientific subjects and those that are committed in everyday life. This work is worth mentioning, since it contains the earliest statements of the modern appeals to 'false' authorities (pseudo-authorities), which are discussed further in this paper. Furthermore, John Locke made a significant contribution to the development of rhetoric and fallacy studies by inventing three main kinds of ad-arguments: 'ad verecundiam' (appeal to authority), 'ad ignorantiam' (appeal to ignorance), and 'ad hominem' (appeal to person) arguments (Hansen 2020). Based on Aristotle's classification of appeals, he identified their hierarchy, discerning 'valid' appeals (appeals to evidence and reason) from 'fallacious' ones (appeals to the speaker's authority and the audience's ignorance) (Longaker 2014). This approach to argumentation provided a basis for modern studies on the criteria used for fallacy identification (Stapleton 2001),

which include arguments supported by claim, reason and evidence; conclusions made from claim and reason; recognition of opposition; and refutation (Khoiri and Widiati 2017).

Overall, it should be noted that fallacies can be broadly put into two main categories: 'formal' fallacies (identified by argument's form) and 'informal' fallacies (identified through the analysis of the argument's content). The latter category may also exploit the emotional weaknesses of the audience and, thus, is often used to enhance the emotional effect of information and manipulate public opinion, which is a central issue in our work. Copi et al. (2018) developed this category by presenting eighteen informal fallacies, including the following three fallacies:

1. 'Argumentum ad verecundiam'—appeals to people that may have no expertise in the given area or 'pseudo-authorities';
2. 'Argumentum ad numerum'—appeals based on the number of people who hold a particular belief or 'pseudo-statistics';
3. 'Argumentum ad populum'—appeals to popular opinion rather than authority or 'stereotypes'.

These three fallacies identified by Copi et al. (2018) serve as a basis for our further classification of logical fallacies (identification with pseudo-authorities, pseudo-statistics and stereotypes) used in the mainstream media coverage of the COVID-19 vaccines.

### 2.2. Digital Pseudo-Identification through the Lens of Lippmann's Theory of Public Opinion and Agenda Setting Theory

With the recent information crisis, the issues of the media and government relationship have become of special interest to many scholars. Early works related to these issues date back to John Stuart Mill's '*On Liberty*', which emphasized absolute press freedom and independence from the state and laid the foundation for our modern understanding of the news media as a 'watchdog' of the state. This libertarian theory, or the free press theory, is in contrast to the authoritarian approach to the media, according to which the role of the state is to control the press in order to protect the interests of society. In the authoritarian theory, there is no feedback allowed from the public, whereas in the Soviet media theory, there is two-way communication and, at the same time, the whole control of the press is under the dictatorship of the country's leader. The social responsibility theory lies between these two approaches and allows the press to have total freedom, but its content should be discussed in a public panel. Another approach, developed by Gramsci, highlights the role of media 'as a key apparatus of the state to produce hegemony' (Yüksel 2013). The 'propaganda model', by Herman and Chomsky ([1988] 2008), considers the media to be dominated by political and business elites, who use the press to 'manufacture consent' in mass public opinion. For the 'indexing approach', elite disagreement is central: if elites agree with each other, news coverage will reflect this consensus; in case they disagree, media are free to cover the range of their opinions but should not go beyond it. Nevertheless, it was Walter Lippmann ([1929] 2021) who first noticed the tendency of the media to serve the state and shape public opinion. His notable book '*Public Opinion*' (Lippmann [1929] 2021) provides a broader lens through which to study the current information crisis caused by COVID-19 and, in particular, the concept of digital pseudo-identification as a means of public opinion manipulation.

According to Lippmann ([1929] 2021), each person has a different perception of reality and social events based on the stereotypes which he or she has. These individual stereotypes, or the pictures inside the heads of these human beings, are their public opinions. People construct their own reality which is, in fact, their own subjective representation of the actual environment. This 'pseudo-environment' is an accumulation of their individual stereotypes or subjective and distorted images of the external world that often mislead them in their judgments. As a result, by attaching emotions to these images, which often do not coincide with reality, people form symbolic pictures of facts on which they base their political actions and public opinions (DeCesare 2012). Thus, fictions become a part of human interactions. Lippmann's ([1929] 2021) ideas refer to the current media crisis when

mass media transmit biased, emotionally charged news stories that often have no relation to scientific knowledge and real facts (Teneva 2021). By embedding emotional elements into news reports, journalists create an illusion that the readers share their opinions and feelings. Hence, modern news-making becomes a process of identification of journalists' opinions with personal beliefs and feelings.

Kenneth Burke (1969), in his famous book '*A Rhetoric of Motives*', considers identification as a key principle of communication that is more important than persuasion. We are all different from each other. In order to overcome this division and become a member of society, we persuade others by getting them to identify with our ways and by speaking their language (Andres 1992). Thus, we are 'both joined and separate, at once a distinct substance and consubstantial with another' (Burke 1969, p. 21). In this regard, identification may be viewed as a general communication principle of identifying the author's opinion with the readers' opinions to make his or her viewpoint acceptable to them and thereby persuade them by using 'valid', verifiable and accurate information (Teneva 2021). Following Burke's theory, Hess (2014) studied how digital devices fundamentally altered the nature of identity. He defined digital identification as a process of technological unconscious consubstantiality, through which users are provided and believe in information and argument based upon their digital substance. Zamparutti (2023) examined how consubstantiality is constructed rhetorically through mass usage of terminology, which has implications for understanding societal connectivity in times of stress and trauma. Thus, the circulation of knowledge in the modern media serves as an echo chamber of personal desire and opinion rather than providing users with diverse perspectives, which corresponds to the abovementioned Lippmann's ([1929] 2021) ideas.

It is worth noting that prior studies considered identification as not wholly deliberate but also semiconscious. For instance, Quigley (1998) viewed Burke's identification within the context of the understanding of language as one way of acting in the world, revealing that in the process of identification, a speaker may use language and other symbols without being fully aware of doing so. Thus, Burke's approach to identification suggests that we should consider the impact of those messages that we do not fully intend to send. According to a study by Kuchel and Rowland (2023), rhetoric is a powerful tool for facilitating an open, two-way exchange of ideas, but it can also be used to confuse and mislead an audience. This idea is significant to the present study which views 'false' identification or 'pseudo-identification' as a type of fallacy that is not recognized by journalists themselves and, therefore, is used unintentionally in vaccine reporting. Regarding the concept of pseudo-identification, prior studies defined it as an explicit and realized falsehood within a cultural context, including false consciousness, implicit attitudes and latent effects (Xiang 2011). In contrast to this approach, the present study views digital pseudo-identification not as a complete falsehood but as inaccurate information that is opinion-based and lacking in factual evidence.

Lippmann ([1929] 2021) reflected on how the media can mislead an audience and serve political interests (Arnold-Forster 2023). Based on his theory of public opinion, McCombs and Shaw (1972) developed agenda setting theory, which highlights the media's influence on public opinion through emphasizing certain agendas and increasing the amount of its coverage in the news. Providing constant and repetitive reporting influences public opinion and moves people to act based on the limited information accessible to them. This limited judgment of an audience concerning events creates a distorted image of the world, thus contributing to the spread of online misinformation. Therefore, Lippmann's ([1929] 2021) theory of public opinion and agenda setting theory are closely related to the present research which views logical fallacies in the mainstream media coverage of the COVID-19 vaccines as a means of manipulating public opinion and propagating online misinformation.

### 2.3. Coverage of the COVID-19 Vaccines in the Mainstream Media

During the first and second waves of the COVID-19 pandemic, journalists served as mediators between the government and the public, taking responsibility for coverage of the vaccines (Perreault and Perreault 2021). The rapid rollout of the world's first registered vaccines against COVID-19 caused skepticism and hesitancy concerning their effectiveness such that some scientists started to view them as health diplomacy tools aimed at enhancing the image of their producing countries rather than as a means of disease prevention (Vargina 2020; Giusti and Ambrosetti 2023). It is worth noting that scholars distinguished two main approaches to developing and promoting the COVID-19 vaccines: vaccine diplomacy and vaccine nationalism (Kirgizov-Barskii and Morozov 2022).

The term 'vaccine diplomacy', conceptualized by Peter Hotez (2014), is neutral by nature. According to Liu et al. (2022), it is a means of achieving national security and an indispensable part of international cooperation in biosecurity. From this stance, vaccine diplomacy is considered as a type of 'corporate diplomacy'—a term that refers to activities related to establishing and maintaining cordial and cooperative relationships either among firms or between firms and national governments, with the aim of pursuing industrial and commercial policies of governments (Strange 1992). As for 'vaccine nationalism', it can be described as putting the interest of a single nation first, above others, for economic or security reasons, when each vaccine-producing country seeks to secure vaccines for its own population and signs deals with pharmaceutical companies directly, limiting the stock available to others. Because of vaccine nationalism, media coverage of COVID-19 has become highly politicized, which contributes to escalating political confrontations between Western and non-Western countries (Kirgizov-Barskii and Morozov 2022).

Recent studies on COVID-19 have revealed misinformation and political slants in the vaccine coverage. For instance, Hart et al. (2020) pinpointed politicization and polarization in COVID-19 news in US newspapers and televised network news. Chipidza et al. (2022) also revealed that COVID-19 coverage was not predominantly health-related, which allowed researchers to identify misinformation both in traditional and social media. Kim (2021) and Ng (2021) examined how the issue of misinformation concerning COVID-19 vaccines affected anti-Chinese and anti-Asian sentiments. Christensen et al. (2022) conducted a thorough analysis of articles on COVID-19 vaccines, concluding that although the mainstream online media were positively polarized towards the vaccines, the coverage of some vaccines was negative. For instance, the 'Oxford-AstraZeneca' vaccine garnered largely negative coverage compared to the 'Johnson and Johnson' vaccine. Notably, Dahlstrom (2021) pointed out that vaccine storytelling frequently contained personal testimonies that contributed positively or negatively to vaccine's images, thus increasing the spread of scientific misinformation. Soares and Recuero (2021) also found that mainstream media coverage can promote the spread of misinformation about COVID-19 when journalists skip a stage of information processing and reproduce false or misleading information. Thus, prior studies have detected misinformation about COVID-19 vaccines in the mainstream media.

## 3. Research Aims, Hypothesis and Methodology

### 3.1. Research Aims

The aim of the present research was twofold: first, to provide insight into the understanding of the concept of digital pseudo-identification as a type of logical fallacy employed in the manipulation of public opinion and propagating online misinformation in the mainstream media and, second, to show its close relationship to Walter Lippmann's and Kenneth Burke's theories, highlighting the relevance of their views in the post-truth era, when the boundaries of what information can be labeled as 'true' and 'false' are blurred. To achieve this aim, the following objectives were set:

1.   To consider the concept of digital pseudo-identification as a type of logical fallacy within the framework of Lippmann's theory of public opinion and to discern it from Burke's concept of identification;
2.   To detect the use of pseudo-identification and measure its frequency in news articles of two major British and American mainstream media sources during the examined period;
3.   To classify the types of pseudo-identification and analyze their role in the vaccine coverage;
4.   To describe the ideological language means deployed in creating distorted representations of the COVID-19 vaccines and in manipulating public opinion.

### 3.2. Research Hypothesis

The study aimed to test the following hypotheses:

1.   News stories about the COVID-19 vaccines contain logical fallacies that are used by journalists unintentionally as proof of their opinions about the vaccines;
2.   Journalists of the mainstream media sources use similar logical fallacies and language means in the vaccine reporting;
3.   Coverage of the European and non-European COVID-19 vaccines is politicized and contains political bias.

### 3.3. Methodology

To achieve the research aims, the study utilized a mixture of research methods, including computer-aided content analysis, as well as rhetorical and CDA analyses. The research process involved several stages.

Firstly, to collect the material, the study utilized Lexis Newsdesk as a data collection tool. Using the search queries 'Oxford-AstraZeneca', 'Pfizer-BioNTech', 'Sputnik V', 'Sinovac' and 'CoronaVac', it became possible to collect a sample of 400 news articles from the online versions of the two major British and American mainstream media sources, including 200 articles from 'The New York Times' and 200 articles from 'The Guardian', respectively. Four COVID-19 vaccines that were developed during the first and second COVID-19 waves and had geographically different producing countries, including two European vaccines (Britain's 'Oxford-AstraZeneca' and Germany's 'Pfizer-BioNTech') and two non-European vaccines (Russia's 'Sputnik V' and China's 'Sinovac' ('CoronaVac')), were selected to make inferences regarding the political bias in the news coverage. To avoid any sampling bias, the study applied a consecutive sampling method, which is regarded as one of the best nonprobability methods, since it seeks to include all accessible subjects as part of the sample. The research timeline between 11 August 2020 and 31 December 2021 was defined by the official registration of the world's first coronavirus vaccine 'Sputnik V' on 11 August 2020 and the approval of the other three vaccines during the following 16 months.

Secondly, to avoid any bias in the content analysis, the collected material was analyzed automatically with the help of the computer-assisted tools. To analyze the lexical content of the articles, the study applied semantic web technologies (Murzo et al. 2022). The growth in computing power and the increase in availability (Kryltcov et al. 2021) made it possible to perform text mining of the collected material. According to Wu et al. (2019), a content-based approach can be very helpful in detecting misinformation. The underlying assumption of this method is that misinformation may consist of certain keywords or combinations of keywords so that a single post or news story with enough misinformation signals can be classified. With the help of https://www.wordclouds.com (accessed on 29 June 2023), we generated two word clouds from the articles of both papers, which allowed us to visualize and measure the frequency of the top 200 keywords in each paper. The occurrence of these keywords allowed for the identification of scientific and pseudo-scientific types of information in the vaccine coverage. To analyze these types of information, we utilized Nexis Newsdesk as a media intelligence tool that allowed for the measurement and classification of mentions related to scientific evidence (opinions of health experts

and statistics with a specific amount of data) and pseudo-scientific evidence (opinions of 'pseudo-authorities', including celebrities, politicians, anonymous experts and vaccinated people; references to generalized statistics or 'pseudo-statistics'; statements, containing metaphors and overgeneralizations about a particular country, which we commonly referred to as 'stereotypes'). Based on these analytics, we proposed a classification of logical fallacies (types of pseudo-identification).

Finally, to describe these fallacies and their ideological language means, we applied Van Dijk's (2006) approach to the CDA analysis, which considers language as a form of power abuse. This approach suggests that most manipulation takes place by text and form, which allowed the researchers to analyze the fragments of the vaccine discourse in a social context and to detect the specific language means used for public opinion manipulation. These fragments included statements about the selected COVID-19 vaccines made by politicians, celebrities and vaccinated people, as well as journalists themselves. To assess the persuasiveness of these discourse fragments, we utilized rhetorical analysis. which made it possible to analyze the elements of a rhetorical situation, including the text, the journalist, the audience, the purpose and the setting, and we evaluated the role of logical fallacies in creating distorted images of the COVID-19 vaccines and manipulating public opinion in the mainstream media.

## 4. Results and Discussion

Using https://www.wordclouds.com (accessed on 29 June 2023), we generated two word clouds from 200 articles of 'The Guardian' and 200 articles of 'The New York Times' concerning the four COVID-19 vaccines ('Oxford-AstraZeneca', 'Pfizer-BioNTech', 'Sputnik V' and 'Sinovac' ('CoronaVac')). The size and color of each word in the clouds highlight its frequency of occurrence, as shown in Figures 1 and 2.

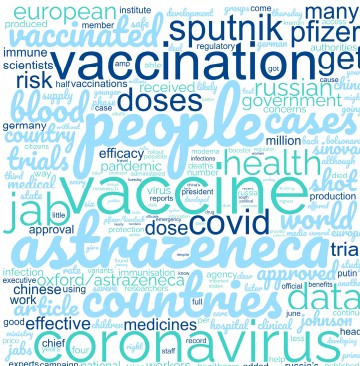

**Figure 1.** The word cloud generated from the articles of 'The Guardian'.

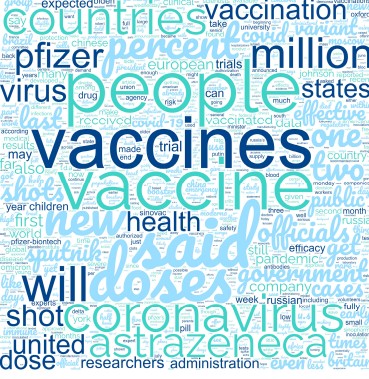

**Figure 2.** The word cloud generated from the articles of 'The New York Times'.

As Figures 1 and 2 show, the articles contained words that are related to scientific and medical information, including 'dose', 'vaccine', 'virus', 'shot', 'health', etc. Notably, the word 'vaccine' appeared to be the most frequent word in the articles of both papers. The names of the vaccines ('Astrazeneca', 'Pfizer', 'Sputnik' and 'Sinovac') and the name of the disease ('coronavirus' or 'COVID') were also among the most frequently used words. These findings illustrate the increasing media attention on the COVID-19 vaccines that was aimed at promoting their images in the eyes of the newsreaders, which is in line with agenda setting theory (McCombs and Shaw 1972). Furthermore, the occurrence of such keywords as 'effective' and 'good' reveals that the vaccines were mainly positively portrayed, which is consistent with similar findings by Malik et al. (2023). Nevertheless, Figures 1 and 2 also illustrate the presence of nonscientific words mainly related to politics, including such words as 'government', 'officials' and 'President'. The occurrence of these words indicates that politics seems to dominate the discussion of the vaccines (DeLay 2021), which underpins the research hypothesis on the politicization of the vaccine coverage.

To detect the types of nonscientific information, we generated a list of the top 200 occurring keywords in each paper based on the two word clouds from Figures 1 and 2. Table 1 presents the ranking and frequency of the most significant keywords that were manually selected and indicated the presence of both scientific and pseudo-scientific information in the articles of 'The Guardian' and 'The New York Times', accordingly.

**Table 1.** The most frequently used words detected in the articles of 'The Guardian' and 'The New York Times'.

| The Guardian | | The New York Times | |
|---|---|---|---|
| **Keywords** | **Mentions** | **Keywords** | **Mentions** |
| 1. vaccine | 1768 | 1. vaccine | 2578 |
| 3. COVID | 442 | 4. doses | 743 |
| 5. Astrazeneca | 403 | 6. coronavirus | 590 |
| 6. doses | 386 | 8. percent | 492 |
| 13. data | 222 | 10. Astrazeneca | 480 |
| 15. Sputnik | 215 | 11. Pfizer | 410 |
| 16. government | 211 | 22. government | 304 |
| 21. Pfizer | 172 | 33. Sputnik | 283 |
| 27. million | 149 | 36. many | 262 |
| 33. many | 133 | 42. Russia | 248 |
| 38. Sinovac | 106 | 52. China | 214 |
| 41. Johnson | 100 | 54. Johnson | 213 |
| 45. President | 91 | 59. Britain | 181 |
| 50. Russia | 88 | 60. researchers | 178 |
| 51. China | 86 | 63. Sinovac | 171 |
| 60. scientists | 73 | 67. President | 169 |
| 70. Germany | 67 | 68. experts | 168 |
| 102. experts | 55 | 74. administration | 161 |
| 111. Putin | 54 | 116. Putin | 123 |
| 113. Bolsonaro | 53 | 169. Biden | 97 |
| 148. researchers | 48 | 180. nearly | 91 |
| 191. several | 42 | 181. federal | 90 |
| 199. Kremlin | 35 | 198. doctors | 81 |

The occurrence of such words as 'scientists', 'experts', 'researchers', 'million', 'data', and 'percent' in Table 1 shows that scientific frames dominated the coverage of the COVID-19 vaccines, whereas mentioning the names of British, American and Russian politicians ('Bolsonaro', 'Johnson', 'Putin' and 'Biden'), political institutions ('Kremlin', 'government' and 'administration'), political titles ('President'), countries ('Russia', 'China' and 'Britain') and quantifiers that do not denote the exact amount of data ('many', 'nearly' and 'several') indicate the presence of nonscientific information in the vaccine coverage. Overall, it is evident from Table 1 that politicians and political organizations were relatively more

popular in the vaccine coverage than health experts. For instance, Boris Johnson was mentioned 100 times in 'The Guardian' and 213 times in 'The New York Times', while the government was mentioned 211 times in 'The Guardian' and 304 times in 'The New York Times' compared to scientists, who appeared only 73 times and 142 times in each paper, respectively. Based on the analytics from the word clouds (Figures 1 and 2), it is revealed that the names of the vaccine-producing countries also spiked in frequency with 248 mentions of Russia, 214 mentions of China compared to 181 mentions of Britain in 'The New York Times', as well as 88 mentions of Russia and 86 mentions of China compared to 67 mentions of Germany in 'The Guardian'. Thus, the analysis of the data from Table 1 allowed for the detection of more media attention on the producing countries of non-European vaccines than European vaccines, which supports our hypothesis concerning political bias in both newspapers. It is also worth mentioning that the surname of the Russian President appeared 123 times, whereas the surname of the US President appeared less often with only 97 mentions in 'The New York Times'. This finding indicates the politicization of the vaccine coverage, which is consistent with recent studies (Hart et al. 2020; Abbas 2022; Teneva and Bykov 2023). Overall, the results from Table 1 reveal the presence of nonscientific information in the vaccine coverage.

To measure and visualize the frequency of the types of information mentioned in relation to the COVID-19 vaccines, the study utilized Nexis Newsdesk as a media intelligence tool. Figures 3 and 4 show the frequency of information detected in the articles of 'The Guardian' and 'The New York Times' on the four COVID-19 vaccines from 11 August 2020 to 31 December 2021.

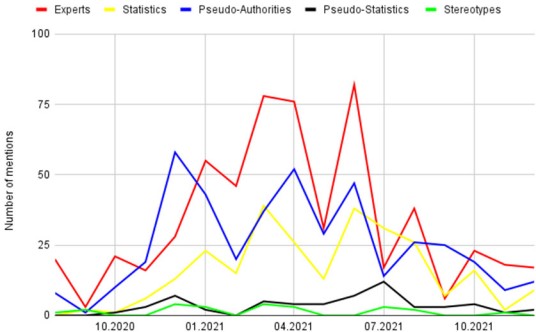

**Figure 3.** Frequency of information detected in the articles of 'The Guardian'.

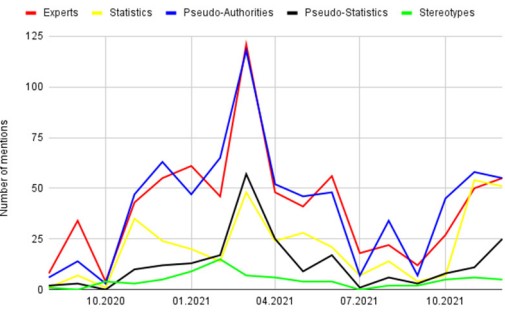

**Figure 4.** Frequency of information detected in the articles of 'The New York Times'.

As is evident from Figures 3 and 4, there were several increases in the frequency of information that were due to the clinical trials and approval of the four selected COVID-19 vaccines between August 2020 and December 2021. Overall, Figures 3 and 4 allowed for the detection of the presence of scientific (1, 2) and nonscientific types of information (3, 4, 5) in the vaccine coverage, including:

1. References to the opinions of health experts, such as reputable scientists, health authorities and scientific journals ('experts');
2. References to statistics ('statistics');

3.　References to the opinions of nonexperts, which included political actors and institutions, celebrities, vaccinated people, and anonymous or 'implicit experts' whose names were not mentioned or hidden from the public eye—'argumentum ad verecundiam' (Copi et al. 2018, p. 120) ('pseudo-authorities');

4.　References to generalized statistics that lack an indication of the specific amount of data—'argumentum ad numerum' ('pseudo-statistics');

5.　Statements containing metaphors or overgeneralizations about a particular vaccine-producing country—'argumentum ad populum' ('stereotypes').

Based on the analytics from Figures 3 and 4, we generated Table 2 which presents the types of mentions and their maximum number of mentions in each paper.

**Table 2.** Types of mentions detected in the articles of 'The Guardian' and 'The New York Times' and their maximum number.

| Types of Mentions | The Guardian | The New York Times |
|:---:|:---:|:---:|
| Experts | 82 | 121 |
| Statistics | 39 | 48 |
| Pseudo-authorities | 58 | 118 |
| Pseudo-statistics | 12 | 57 |
| Stereotypes | 5 | 15 |

A comparison of the data presented in Table 2 reveals the discrepancies in the vaccine coverage of the two major British and American mainstream media sources. The articles of 'The New York Times' contained more references to the opinions of health experts (121 maximum mentions) than pseudo-authorities (118 mentions), as well as more appeals to pseudo-statistics (57 mentions) than statistics (48 mentions). In contrast, the articles of 'The Guardian' contained more references to the opinions of experts (82 mentions) than pseudo-authorities (58 mentions), as well as more appeals to statistics (39 mentions) than pseudo-statistics (12 mentions). References to stereotypes were the least common in both papers, with 5 peak mentions in articles of 'The Guardian' and 15 peak mentions in the articles of 'The New York Times'. Overall, the findings from Tables 1 and 2 show that the articles of 'The New York Times' contained more pseudo-scientific information than the articles of 'The Guardian'. Thus, based on the data from Table 2, we proposed our classification of logical fallacies, which includes identification of journalists' opinions with pseudo-authorities, pseudo-statistics and stereotypes.

*4.1. Identification with Pseudo-Authorities*

When reflecting on factors contributing to the spread of online misinformation, Froehlich (2019) distinguished between 'honest authorities', whose knowledge is based on verifiable knowledge, and 'pseudo-cognitive authorities', who appear to have credibility and expertise but on critical examination fail in these qualities and strive to impose a partisan agenda irrespective of truth, evidence, logic or facts.

The discourse analysis of the news stories on the COVID-19 vaccines revealed that journalists of both papers identified their opinions with the opinions and feelings of pseudo-authorities who do not possess in-depth knowledge and sufficient competence to form fully informed judgments concerning the scientific matters but whose opinions seemed to be more trustworthy and authoritative to the public than the opinions of real experts and scientists. The articles contained frequent references to the opinions of reputable politicians, for example:

1. *'I trust AstraZeneca, I trust the vaccines,' Ursula von der Leyen, the top European Union official, said at a news conference in Brussels* (Cohen 2021).

2. *The prime minister went on: 'But we're working together on the AstraZeneca it's a great vaccine. I have AstraZeneca.'* (Walker and Phillips 2021).

3. *Xi Jinping, China's leader, called it [CoronaVac] a 'global public good'* (Wee 2021).

4. *In France, President Emmanuel Macron talked to Mr. Putin recently about possible deliveries of Sputnik, which Mr. Macron's foreign minister derided as a 'propaganda tool'. [...] Mr. Matovic faced a revolt from his own ministers, [...] succumbing to what his foreign minister, Ivan Korcok, described as a Russian 'tool of hybrid war' that 'casts doubt on work with the European Union'* (Higgins 2021).

Examples № 1–4 contain quotes with the opinions of politicians concerning COVID-19 vaccines. Images of politicians are used as a means of advertising either a person or an issue in the modern media (Dorofeev 2023). Thus, their image is transferred to the vaccines, highlighting either positive or negative vaccine qualities. References to politicians' opinions serve as a means of creating either positive (№ 1–3) or negative (№ 4) vaccine images, increasing the popularity of both the vaccines and the politicians in the news coverage. Example № 4 supports the research hypothesis regarding the political bias in the news coverage. The opinion of the politician steers the readers' minds towards his negative opinion concerning the foreign vaccine and its producing country. This finding is in line with the ideas of Abbas (2022) regarding ideological polarization in mainstream media coverage when journalists highlight the positive qualities of 'our' vaccines, on the one hand, and negative qualities of others' vaccines, on the other hand. Furthermore, examples № 1–4 show that political language contains specific connotations and ideological meanings that encourage the public to act in a way that is favorable to politicians. Emotion-laden words that denote either politicians' approval of the vaccines ('trust', 'support', 'great' and 'safe') or their disapproval ('cast doubt', 'a propaganda tool' and 'a tool of hybrid war') are intended to incline the audience towards positive or negative opinions about the vaccines and prejudicial attitudes, which reflect their emotional and political bias. Thus, emotional elements are typical indicators of public opinion manipulation (Teneva and Bykov 2023).

Recent studies (Brennen et al. 2020; Dahlstrom 2021) have revealed that vaccine narratives containing the personal testimonies of politicians, celebrities and public figures have a significant impact on vaccine attitudes and behaviors, even if they contain inaccurate scientific information. According to Brennen et al. (2020), high-level politicians, celebrities or other prominent public figures produced or spread only 20% of the misinformation in the vaccine coverage, but that misinformation attracted a large majority of all social media engagements. Similarly, DeLay (2021) has highlighted the role of political and social actors in framing science-related policy issues in public discourse and mobilizing support for their position or perspective. These findings recall the concept of social proof, developed by Cialdini (2007), where people copy what other people do. The analysis of the news stories in both papers revealed frequent references to personal testimonies of politicians who were vaccinated or who urged the public to get vaccinated, for example:

5. *PM has first dose, calling experience 'very good, very quick' and urging Britons to get vaccinated* (Walker 2021).

6. *President Biden went out of his way to draw attention to Pfizer-BioNTech's findings on Wednesday, calling them "very, very encouraging. [...] If you get the booster, you're really in good shape," Mr. Biden said* (LaFraniere 2021).

Examples № 5 and 6 illustrate that the personal testimonies of politicians often serve as pseudo-scientific evidence in favor of a vaccine's effectiveness. These testimonies are aimed to prove that if Boris Johnson, or any other politician, is vaccinated with this vaccine and feels well, then it can be regarded as effective and safe. The feelings of the former British Prime Minister after being inoculated with the Astrazeneca vaccine and the opinion of the US President who urges the public to get inoculated with the Prizer-BioNTech vaccine are

provided to promote the effectiveness of these vaccines without providing any scientific evidence. Emotional elements that are embedded into the politicians' statements, including adjectives denoting positive feelings ('encouraging' and 'good') and intensifying adverbs ('very' and 'really'), exert emotional influence on the audience and, thereby, manipulate public opinion (Teneva and Bykov 2023).

The findings of the study reveal frequent mentions of the opinions of celebrities used as 'advertisements' of the COVID-19 vaccines, for instance:

> 7. *British stand-up comedian Lenny Henry says, 'the vaccine [AstraZeneca] does not contain the live virus and is definitely working'* (May 2021).

> 8. *Lionel Messi has helped to obtain 50,000 COVID vaccines from China for an ambitious but controversial plan to inoculate all of South America's football players [. . .]* (Goni 2021).

Examples № 7 and 8 demonstrate that the opinions of celebrities are a way to advertise the effectiveness of the COVID-19 vaccines. Celebrity endorsements contribute to public opinion manipulation and creating a more favorable and positive vaccine image. The credibility of such claims depends not on the competence of celebrities on health issues or provable scientific facts but rather on the popularity of famous people among the audience. The more popularity celebrities have, the more convincing their arguments are likely to be. This finding is congruent with the understanding of the 'post-truth' crisis, where personal opinions have become more 'trustworthy' than facts (Giordani et al. 2021).

The results of the study indicate frequent references to the testimonies of the vaccinated people who are either in favor (№ 9) or against (№ 10–12) the COVID-19 vaccines because of political and health reasons, for example:

> 9. *'I'm more than happy to get the vaccine [AstraZeneca] myself, though I feel it's immaterial' Lewis, 29, architect, London* (Obordo and Guardian readers 2021).

> 10. *'I don't trust the government. There's no way I'm taking the vaccine [Sputnik V],' said one Moscow teacher, who declined to be named* (Beaumont and Harding 2020).

> 11. *'Even right in the middle of this emergency, I have no reason to trade my life or my family's for a Chinese vaccine,' said Nguyen Hoang Vy, a manager for health care operations at a hospital in the city of Ho Chi Minh* (Wee and Lee Myers 2021).

> 12. *Yasmine Cotton, 19, health care assistant and student. 'Now, I just feel extremely worried. Every headache I get I think is this the blood clot? It's terrifying.'* (Blackall 2021).

Examples № 9–12 illustrate citations containing personal opinions of vaccinated people about the COVID-19 vaccines. Emotional vocabulary denoting positive or negative feelings of these people about or after the vaccination ('happy', 'extremely worried' and 'terrifying') is intended to trigger similar emotional responses in the audience and make them identify with the feelings and opinions of these people about the vaccines, thus contributing to public opinion manipulation and creating distorted vaccine images. Examples № 10 and 11 show that vaccines were portrayed in terms of public distrust of the Russian government and anti-China sentiment in Vietnam, which supports the hypothesis about politicization of the vaccine coverage (Hart et al. 2020; Kim 2021; Ng 2021; Abbas 2022; Christensen et al. 2022).

The results of the study also show that the opinions of anonymous or 'implicit' scientific experts, whose names were either not mentioned or hidden from the public eye, were widespread in the vaccine coverage of both papers. In these cases, words with a generic meaning of authority were common, for instance:

> 13. *Observers say the Sputnik V jab is aimed more at sowing political division than fighting coronavirus. [. . .] EU observers say Moscow is deploying Sputnik as another weapon of geopolitical influence.* (Henley 2021).

> 14. *Experts all agree that AstraZeneca is a safe vaccine* (Boseley 2020).

References to the opinions of implicit scientific experts ('observers' and 'experts') whose names were not mentioned in the text of the news stories create an illusion that the provided opinions are credible and trustworthy. They are used to promote either trust (№ 13) or skepticism (№ 14) towards the effectiveness of the COVID-19 vaccines. This finding is in line with the results of a recent study (Teneva 2021), which identified the types of pseudo-authorities in the vaccine coverage, including nonexperts whose names are mentioned ('nominal' pseudo-authorities) and not mentioned ('implicit' pseudo-authorities).

Appeals to journalists' personal opinions are another way to enhance the image of both the vaccine and journalists themselves in the news coverage, for example:

15. *I got my first AstraZeneca shot. The only lasting effect has been a sense of relief. [...] I am very happy to have had the AstraZeneca vaccine* (Butler 2021).

16. *Why I Got the Russian Vaccine* (Kramer 2021).

As is evident from examples № 15 and 16, news stories containing journalists' personal experience of vaccination often serve as 'trustworthy' arguments in favor of the COVID-19 vaccines and are intended to engage the readers emotionally, making them trust journalists without any factual evidence. Thus, the image of journalists becomes a means of advertising the positive qualities of the vaccines. It is a way of disseminating online misinformation by replacing scientific evidence with personal opinions on the vaccine issues.

### 4.2. Identification with Pseudo-Statistics

It goes without saying that information is one of the most valuable resources in the modern world (Matrokhina et al. 2021). However, distinguishing 'information' from 'pseudo-information' has become a challenging task in the post-truth era, when facts are often confused with opinions and beliefs (González-Méijome 2017; Kim and de Zúñiga 2021). Lippmann ([1929] 2021) also reflected on the ways in which information is chosen in the media and organized to serve someone's interests. According to his theory, people are impressed by those facts which fit their philosophy. In other words, facts do not convince the audience if they are contrary to their views or stereotypes, which contributes to the spread of personal opinions and beliefs, as well as misinformation.

The findings of the study reveal that apart from providing scientific data, journalists frequently utilized references to statistical overgeneralizations or 'hasty generalizations', which suggests making conclusions without providing accurate factual evidence. Statements containing these pseudo-statistics were provided to reinforce journalists' arguments about the COVID-19 vaccines, for example:

17. *Many Russian liberals reflexively rejected the vaccine [Sputnik V] because of its association with the Kremlin* (Troianovski 2020a).

18. *Much of the world is looking to AstraZeneca in part because it has set more ambitious manufacturing targets than other Western vaccine makers* (Mueller and Robbins 2020).

19. *Much of Latin America has relied on the Chinese and Russian vaccines, and on AstraZeneca* (Nolen 2021).

20. *Several million pediatric doses of Pfizer-BioNTech's coronavirus vaccine should be available in the next few days* (LaFraniere 2021).

Examples № 17–20 illustrate statistical overgeneralizations either about the number of people who are either in favor or against the vaccines or about the amount of the vaccine supply. Quantifiers, which have a meaning of an unspecified large amount ('many', 'much', 'several', etc.), and collective nouns ('millions', 'thousands', etc.) are typical indicators of such pseudo-statistics. Using this vocabulary contributes to promoting (№ 18–20) or discrediting (№ 17) the vaccines and creates their distorted images. Notably, this kind of pseudo-scientific evidence is used not only as factual information but also as a means of emotional manipulation, since it lacks any accurate scientific data. Large numbers are aimed at increasing the emotionality of the arguments in the eyes of the newsreaders, which

is a way to manipulate public opinion. Examples № 17 and 18 demonstrate the political and economic reasons behind choosing or rejecting the vaccines, which indicates political framing of the vaccines and supports our research hypothesis about the political bias in both mainstream media.

The results of the study reveal frequent references to generalized statistics about the vaccine preferences of particular nations, for example:

21. *Polls are finding Americans increasingly wary of accepting a coronavirus vaccine [AstraZeneca]* (Grady et al. 2020).

22. *Many Chinese had also been hesitant to get the shots [of 'CoronaVac'], in part because of past scandals involving Chinese-made vaccines* (Qin and Chang Chien 2021).

23. *Mongolians have also expressed a preference for Russia's Sputnik vaccine* (Stevenson 2021).

Examples № 21–23 demonstrate journalists' overgeneralizations about the vaccine preferences of each nation, which lack any scientific evidence and data. The vaccine preferences are portrayed in terms of political interests of the corresponding country. The exact number of people 'for' (№ 23) or 'against' (№ 21 and 22) is not provided, which contributes to public opinion manipulation and creates either positive (№ 23) or negative (№ 21 and 22) vaccine images.

### 4.3. Identification with Stereotypes

The crucial role of stereotypes has been recognized by many scholars. Lippmann ([1929] 2021) first noticed the tendency of journalists to generalize about other people based on stereotypes or popular opinion. In the post-information society when information is transmitted and processed faster than a human thought (Vinogradova et al. 2020), the processes of perception of information have changed a lot. Using stereotypes simplifies our perception of the world and directs out attention towards particular information (Sherman 2022). Nevertheless, they can influence our decisions, create cognitive bias towards gender, nation, race, etc., and even lead to collective self-deception.

The results of the study indicate the presence of statements that contain metaphors and overgeneralizations about the vaccine-producing countries, which we referred to as the fallacy of stereotyping. Metaphors play a significant role in stereotype formation. They create new realities (Kövecses 2018), have the power to shape our perception of the world and can, to some extent, influence our actions (Carter 2021). In order to interpret metaphors and make inferences about their meaning, the context should be taken into account (Pushmina and Carter 2021). Recent studies (Lahlou and Rahim 2022; Teneva and Bykov 2023) have found that conceptual metaphors of war dominate vaccine discourse. The findings of the discourse analysis of the news stories show that war metaphors were very common in both papers. Journalists often referred to stereotypes related to the Soviet Union and the Cold War, for example:

24. *It is no accident that Russia has named its vaccine Sputnik V, harking back to the Soviet satellite sent into orbit in 1957 amid fierce competition with the US. For Russia, providing the first solution to a pandemic that has affected every corner of the world would be seen as a confirmation that the country's scientific brains are still among the world's best* (Walker 2020).

25. *Malte Thiessen, a historian of immunization, told German media that the vaccine was seen as a huge opportunity in Russia for it to polish up its image abroad. 'Just the name Sputnik is a first-class piece of propaganda,' he said.* (Connolly 2021).

According to Lippmann ([1929] 2021), we perceive reality through a 'stereotyped' vision. Mentioning the 'Sputnik V' vaccine in examples № 24 and 25 as a metaphorical reference to the stereotypes about the communist era and the Soviet Union may have a double meaning: as a negative symbol of Russia's propaganda and aggressive foreign policy and as a positive symbol of Russian medical breakthroughs in the vaccine race.

Example № 24 shows the stereotypes about Russia's first artificial earth satellite and the space race during the Cold War, which creates a positive image of Russia as the country in which 'scientific brains are still among the world's best', while in example № 25, the same stereotypes of the Soviet Union reflect negative attitudes and political bias towards the producing country of the vaccine, which is regarded as 'a first-class piece of propaganda'. Thus, this finding also supports the research hypothesis on the political bias in the vaccine coverage.

The findings reveal conceptualizations of the COVID-19 vaccines as 'weapons', for example:

> 26. *'Our Sputnik V is unpretentious and reliable, like the Kalashnikov rifle,' the state television host Dmitri Kiselyov said on his show month* (Troianovski 2020b).

> 27. *'I would not get AstraZeneca because that would be like playing Russian roulette.'* (Cohen 2021).

Gun-related metaphors contribute to either discrediting (№ 26) or enhancing (№ 27) the image of the vaccines and their producing countries. On the one hand, in example № 26 comparing 'Sputnik V' to the most famous Russian weapon, a Kalashnikov rifle, this evokes positive stereotypes of the Soviet Union and its reliable product, thus inclining the audience towards the idea of the safety and high quality of the Russian vaccine. On the other hand, this metaphor evokes negative stereotypes of Russia as a country that boasts its power and weapons and poses a threat to the rest of the world. In example № 27, the 'Oxford-AstraZeneca' vaccine is compared to a potentially lethal and dangerous game, which involves the use of a gun—'Russian roulette'. The feelings of fear and danger that are associated with this game are extended towards the British vaccine, creating its negative image. These findings confirm that metaphorical language is a powerful tool for manipulating public opinion (Van Dijk 1998).

The findings also show that stereotypes related to the quality of the national medicine were also widespread in the vaccine coverage, for instance:

> 28. *Russia has plenty of world-class scientists, and the Gamaleya Institute claims to have had a head start* (Twigg 2020).

> 29. *Anti-China sentiment runs high in Vietnam, but the country accepted a donation of 500,000 doses of Sinopharm in June, causing a backlash among citizens who said they did not trust the quality of Chinese shots* (Wee and Lee Myers 2021).

> 30. *Some doctors and activists have put forward proposals to increase the delivery worldwide of vaccines produced in the West. These calls are well-intentioned, but they, too, assume that vaccines from Western countries are the only ones worth having—and waiting for* (Prabhala and Yoke Ling 2021).

Examples № 28–30 illustrate positive and negative stereotypes about the quality of Russian, Chinese and Western medicine. This vaccine framing may have an unintentional effect, producing specific prejudice against the vaccine-producing countries. Thus, example № 28 contains exaggerations about the professionalism of Russian scientists. Emotionally charged vocabulary, including the word 'plenty' as a quantifier, which means 'a large quantity', and the adjective phrase 'world-class', which denotes 'being of the highest degree of excellence in the world', is intended to create a positive image of both the vaccine-producing country and its health experts. Example № 29 illustrates the political prejudice against the Chinese vaccine, where China is viewed as a culprit in the pandemic (Kim 2021), whereas in example № 30, the reference to 'implicit' experts ('some doctors and activists') reveals the journalist's own preference in favor of Western vaccines. The indefinite pronoun 'some' is used to hide the source of information. These exaggerations, prejudices and stereotypes contribute to public opinion manipulation in the vaccine coverage.

## 5. Conclusions

To summarize, this study aimed to consider the concept of digital pseudo-identification as a tool for manipulating public opinion concerning the COVID-19 vaccines and disseminating online misinformation. The findings of the study confirm that, apart from scientific data, the vaccine coverage of the two major British and American mainstream media sources contained information that was pseudo-scientific and mainly related to politics. This information was used by journalists unintentionally as 'false' arguments to support their claims about the vaccines, which supports the research hypothesis about the presence of logical fallacies in the mainstream media coverage of the COVID-19 vaccines. The computer-aided content analysis of the collected data revealed that the journalists of 'The Guardian' and 'The New York Times' used similar logical fallacies, including the opinions of pseudo-authorities and references to pseudo-statistics and stereotypes. Using these fallacies creates distorted images of the COVID-19 vaccines, which can manipulate public opinion and lead to false or invalid conclusions concerning the vaccines' effectiveness, based on faulty logic, pseudo-scientific evidence or political bias. Thus, using logical fallacies in vaccine coverage poses a serious threat to the credibility of the media and science. Detecting these fallacies is a challenging task both for scientists and professional journalists, since they are often embedded in the rhetorical patterns that obscure the logical connections between statements. Therefore, the present research intends to help communications platforms, journalists and fact-checkers worldwide improve their classifications of 'false' information and detect logical fallacies in order to combat the spread of online misinformation about COVID-19.

Notably, the findings of the content analysis revealed discrepancies in the vaccine coverage: the articles from 'The New York Times' contained more logical fallacies than the articles of 'The Guardian', which means that 'The New York Times' is more vulnerable to online misinformation than 'The Guardian'. The results also support the research hypothesis on the political bias in both papers, with relatively more positive coverage of domestic (European) vaccines than foreign (non-European) ones. Frequent mentions of famous political actors and institutions in relation to the COVID-19 vaccines were aimed at either promoting or discrediting the COVID-19 vaccines, which contributed to public opinion manipulation. These findings show that the mainstream media coverage of the COVID-19 vaccines is highly politicized, proving that 'the propaganda model still works well' (Herman and Chomsky [1988] 2008).

The results of the discourse analysis show similarity in the use of ideological language means by the journalists of both papers. These means include common nouns with a generic meaning of authority, collective nouns and quantifiers that denote large numbers, indefinite pronouns that do not refer to specific persons or things, and emotionally charged vocabulary. Expressive language means, such as gun-related metaphors, were used to induce negative feelings, such as skepticism and fear towards the vaccines, highlighting the idea that the vaccines were regarded as 'weapons' in the news reporting. Interestingly, the same stereotypes of the Soviet Union were used to show both positive (a reference to 'a Kalashnikov rifle') and negative attitudes (a reference to Soviet propaganda during the Cold War) towards the Russian vaccine in both media, which reveals the dual nature of stereotypes and their significant role in interpretation of the information (Lippmann [1929] 2021).

Overall, identification of logical fallacies helps understand how an argument may be incomplete or 'false' and develop critical thinking and media literacy skills that are necessary to combat online misinformation. Therefore, the present research opens us perspectives for further education on media literacy and misinformation detection. It also reveals a shift in modern journalism from evidence-based reporting to opinion-based reporting when personal opinions and beliefs have become more prevalent and 'reliable' than scientific evidence and facts, which confirms the relevance of the ideas of Walter Lippmann ([1929] 2021) in the post-truth era.

## 6. Limitations of the Study and Further Research

There are some significant limitations of the study and suggestions for future research. The findings of the study are not applicable to all vaccine news coverage and relate only to the two major mainstream media sources (The Guardian and The New York Times). Although the present research makes some effort to propose a new understanding of pseudo-identification as a type of fallacy that contributes to misinformation, we are aware that references to nonauthorities and nonstatistical descriptive reporting might be referred to the common business practices of media and television production during the COVID-19 pandemic. As we mentioned earlier in the paper, appeals to reason, authority and emotions can be regarded as a means of both persuasion and manipulation. Thus, to avoid any ambiguity in interpreting the qualitative data, references to pseudo-authorities, pseudo-statistics and stereotypes are considered as a means of manipulation through a lens of agenda setting theory, which explains how increasing the amount of coverage of both politicians, celebrities and other nonexperts in the vaccine coverage impacts positive or negative images of COVID-19 vaccines and manipulates public opinion. Logical fallacies may sound convincing, but they lack evidence that supports their claim, which is in line with previous studies (González-Méijome 2017; Dahlstrom 2021; DeLay 2021; Kim and de Zúñiga 2021; Kuchel and Rowland 2023) that considered personal testimonies, uncertain facts and stereotypes as pseudo-scientific information or logical fallacies which lack scientific evidence. Therefore, future research might consider ways to distinguish identification, which functions as a mechanism of persuasion from pseudo-identification as a tool for public opinion manipulation within a broader social and cultural context in humanities and social sciences.

**Funding:** This research received no external funding.

**Institutional Review Board Statement:** Not applicable.

**Informed Consent Statement:** Not applicable.

**Data Availability Statement:** The data presented in this study are available upon request from the corresponding author.

**Conflicts of Interest:** The author declares no conflict of interest.

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
