# Peer review of "Digital Pseudo-Identification in the Post-Truth Era: Exploring Logical Fallacies in the Mainstream Media Coverage of the COVID-19 Vaccines"

_socsci, doi:10.3390/socsci12080457_

Round 1

Reviewer 1 Report

I have read this article with interest. Before considering this manuscript for publication, the authors should improve a series of areas that I indicate below:

When using the concept of Deep meditization, the authors should consider the works of A. Hepp to enrich their theoretical construction.

In the introductory section, the objective should be specified more specifically. Now it is very general.

In section 2.2 it is necessary to incorporate more recent authors since pseudo-identification in the digital environment is addressed. In addition, more previous research on the news coverage of COVID-19 vaccines should be incorporated into the literature review. Now this section is poor.

Hypothesis 1 is incorrect. This research is based on an analysis of newspaper news content. Therefore, it cannot generate results or statements about the perceptions of citizens. For that you would need to conduct a survey or resort to interrogation methods. The authors must completely reformulate hypothesis 1 since now it cannot be demonstrated with the data generated by this investigation.

On the other hand, hypothesis 2 should also be improved to become clearer and, above all, more easily verifiable with the data generated by this study.

The authors should justify why they have chosen the two newspapers studied for this research. Why these ones and not others? Why only two newspapers and not more?

In addition, they should also justify why they have used the search queries and not others. There are other vaccines that have not been considered as search queries, for example, Pfizer/BioNTech or Moderna. This can bias the results.

The three strategies used in the CDA (identification of journalists' opinions with pseudo-authorities, pseudo-statistics and stereotypes) should be better explained. The authors should explain how they have empirically applied these strategies, that is, they should explain how they have been operationalized for their analysis.

Quantitative content analysis is very limited. It offers little data. This should be explained. Why? Most of the results focus on qualitative analysis. Why?

In the discussion, the authors state that vaccine hesitancy is largely fueled by online mis- and disinformation which occurs through the use of digital pseudo-identification. In my opinion, the results do not support this strong statement. The authors should qualify it. First, not alluding to disinformation and second, reducing its forcefulness. I think that now they accuse journalists of promoting mis- and disinformation in a general way with the information about vaccines, which is not true and cannot be generalized from qualitative data. It is necessary to be more prudent in these types of statements to maintain scientific rigor.

The authors should also make it clear in the discussion that the results are not generalizable to all vaccine news coverage or even to all British and American newspapers. Since most of the results come from a qualitative analysis, this affects representativeness and the results cannot be extrapolated beyond the two newspapers studied.

In the discussion section, the results obtained are not critically compared with the findings of previous research. Consequently, there is no discussion, something that must be corrected. Furthermore, the authors neither indicate the limitations of this research nor do they critically reflect on the implications of these limitations. This should also be corrected by the authors.

In the conclusions, the authors state (line 685) that the coverage of the COVID-19 vaccines has a profound effect on the level of vaccine trust and vaccination. This statement is not derived from your data or your findings. Therefore, the authors have two options: delete it or cite previous research that offers data on it. In general, the first paragraph of the conclusions is inadequate for this very reason and needs to be completely rewritten. Their claims are not derived from the study conducted.

Author Response

Dear Reviewer

Thank you very much for your valuable comments. We tried to pay attention to all of them, including:

  1. Introduction was revised and the objective specified. The works of Hepp were added and considered.
  2. We changed the research hypothesis and excluded information about the perceptions of the audience
  3. We updated literature into Section 2.2. and added some recent studies on identification in the digital era. It should be noted that there is a limited number of scholarly works related to pseudo-identification.
  4. We explained the reason for choosing the four vaccines and two newspapers in Methodology section.
  5. We have better explained three logical fallacies and expanded quantitative content analysis.
  6. We tried to reformulate the statements in the CDA to make the statements less forceful and stated that journalists use these fallacies unintentionally so that not to accuse them in the spread of misinformation.
  7. The section Findings and Discussion was updated with critical analysis of literature.
  8. Conclusion is re-written based on the findings of the study.

Reviewer 2 Report

This is a promising manuscript. It addresses a very important topic: the failures of mainstream liberal journalism to establish its own credibility in the midst of a global pandemic and “infodemic.” The author(s) clearly have a mastery of foundational texts in the fields of persuasion and propaganda, and parts of this paper (the literature review in particular) were a genuine pleasure to read.

However, there are several significant gaps in the paper’s theory, methodology, and analysis which require attention before I believe this paper will be ready for publication.

First, there is the relatively simple matter of claiming intent on the part of journalists to manipulate their audience. This lacks warrant. It seems far more likely that most journalists themselves—who are generally not trained in public health, statistics, or rhetoric—simply do not appreciate the difference between a reasoned analysis of vaccine risk/reward and the kinds of argument-by-pseudo-identification the author(s) are addressing. At any rate, it is unknown, and I believe the manuscript should be revised so as not to imply nefarious intent where none can be proven. This is not a trivial issue, as it lies at the root of the difference between misinformation on one hand and disinformation and malinformation on the other.

Relatedly, the paper’s core argument misapplies Lippmann’s model of stereotyping (which is descriptive) by applying normative values to the presence or absence of stereotypes. I believe this is a misunderstanding of Lippman and the function of stereotype. Take as an example: Even the average, educated reader of the New York Times or Guardian is neither a statistician, virologist, nor a public health expert. Therefore, these readers’ trust in public health authorities is itself based on a stereotype of those authorities’ impartiality, intelligence, and sense of public interest. True, this stereotype is more accurate than if they had stereotyped the public health authority as, for example, an agent of foreign conspiracy. But it is nevertheless a stereotype in the Lippmannian sense. The author(s) appear to make the normative critique that journalists can escape this dynamic, but I believe that Lippmann’s own work suggests that this is an inevitable feature of mass-mediated society. If the author(s) wish to argue that this isn’t the case, that argument might be made. But in either case it should be addressed.

Also related: the author(s) should address the roles of framing and agenda-setting and how they might attenuate some of the author(s) claims. There are, of course, a variety of “beats” in major publications such as the New York Times and Guardian. There is a public health beat, a politics beat, an entertainment beat, and so on. Given the ubiquitous nature of a pandemic, COVID-19 would have affected news across all these beats and more. Non-authorities and non-statistical descriptive reporting are not inappropriate on some journalistic beats. So, for example, a story about celebrities taking their vaccines might indeed belong to the category of “identification with pseudo-authorities.” But it might just as well refer neutrally to the business practices of television and movie production during the COVID-19 pandemic. Such stories may or may not implicitly endorse a celebrity’s choice, thus making such stories weak support for the author(s) thesis. Was there a mechanism for avoiding these false positives? The manuscript should specify if there was or attenuate if there was not.

Similarly, public opinion itself, and politicians’ statements and actions, are newsworthy, and vaccination itself was fodder for political and “culture war” conflict. Threfore a story addressing these angles would need to report on public opinion, but would not necessarily endorse the opinion being reported on. The Times reported on COVID denialism by the Trump White House, but the framing of these stories was critical, not endorsing. Byut b the methodology described in the manuscript such stories could fall into the category of “identification with pseudo-authorities.” Again, if methodological steps were taken to avoid such false positives, the manuscript should reflect that. If such methodological steps were not taken, then some of the manuscript’s claims should be attenuated. It may be worthwhile to revisit coding in like of framing and agenda-setting, and perhaps expand it beyond the categories of “scientific” and “non-scientific” evidence.

Again, this is a promising manuscript, and I hope these notes help. Best of luck to the author or authors.

Author Response

Dear Reviewer,

Thank you for providing valuable comments on the manuscript. We tried to pay attention to each of them and made major changes in the manuscript, including:

  1. Following your comments on stereotypes and other fallacies, to avoid any ambiguity in interpreting qualitative data, we revised the paper and considered logical fallacies as a means of misinformation that is spread unintentionally so that journalists do not recognize these fallacies themselves. 
  2. We tried to attenuate some claims in the CDA , explaining that logical fallacies can both promote or discredit the image of the vaccines. However, even in positive cases, they rely on inaccurate or opinion-based information that lacks scientific evidence, which indicates misinformation.
  3. We added Agenda Setting theory as a theoretical framework of the study to attenuate some claims
  4. We explained the frequent references to non-authorities and pseudo-statistics as a way of drawing attention to the vaccines and shaping public opinion about them.

Thank you for consideration of our manuscript and your thorough analysis of the study!

Reviewer 3 Report

The impact of misinformation on vaccine hesitancy is an important and timely topic, so I’m glad to see that it is being explored! The goals of this study were well-articulated and the methods were clearly described. I do think that the Introduction & Theoretical Context sections would benefit from more focus on the spread of (mis)information in the digital age. For example, misinformation, disinformation, and Fake News are never defined. These are key concepts for the article and should be defined in the introduction. I also suggest deleting the sentence in lines 23-24 and reworking the first paragraph to more broadly introduce the topic of misinformation in journalism, then narrow the focus to your specific context of news on the Covid vaccines.

I do not have any major concerns with this piece and am looking forward to seeing it published. I am including some suggestions below to improve clarity/readability.

- In line 68 delete “Aristotle”

- In line 89 the authors describe “ad hominem” as appeal to motive, but it is appeal to the person

- Delete “This” in line 93

- Line 238 should say “Our study aims to test the following two hypotheses” rather than “to verify.” This wording sounds less biased

- The authors provide clear definitions of pseudo-statistics, pseudo-authority, and stereotypes in section 3.3 but I wish they were defined earlier. They should certainly be defined before introducing the research aims and hypotheses

- Lines 600-601 should read “in the vaccine race.”

- Delete “a” before “proof” in line 677

- The final sentence (lines 698-701) should be reworded to improve readability

Author Response

Dear Reviewer,

Thank you for providing valuable comments. We tried to revise our paper based on them and made major changes, including:

  1. We revised Introduction with more focus on the concept of misinformation and the reasons for its spread.
  2. We made corrections of all the statements mentioned in the review.
  3. We reformulated the final sentence and conclusion.

Thank you for consideration of our manuscript for publication and your important remarks.

Round 2

Reviewer 1 Report

I thank the authors for the effort they have made to improve their manuscript. However, in my opinion there are still several problems and weaknesses that need to be resolved before considering publication of this research. They are the following:

In the introduction, authors should consider references on the perception of citizens about disinformation during COVID-19 such as: https://doi.org/10.7195/ri14.v21i1.1988 and https://doi.org/10.37016/ mr-2020-024

The hypothesis continues to pose problems. First, it is very general and imprecise. This makes it difficult for it to be verified correctly. Second, it makes claims that cannot be proven with data, such as the statement about shaping public opinion and disseminating online misinformation. The investigation analyzes the content of two newspapers. No data can be obtained from this investigation that allows us to infer or demonstrate these statements. It is not possible to say “shaping public opinion” in the hypothesis why analyzing two newspapers does not have data to verify this. Authors must remove the final part of their hypothesis. I also recommend that the authors must rewrite it using two or more hypotheses so that they are more concrete (and closely linked to the available data) so that they can be correctly verified.

The methodology affirms "this paper examines news stories as a means of shaping public opinion". This is incorrect. If the content of two newspapers is analyzed, this cannot be affirmed in the methodology. Authors should remove this and rewrite the section accordingly without making any claims about “shaping public opinion”.

In general, authors should remove all references to what their results allow to know as "shaping public opinion" in relation to the topic. This is not true with the available data.

In my opinion, the Conclusions section should come before the Limitations of the study and further research section. The current order should be reversed. In this way, the structure of the manuscript will be more coherent.

In the Conclusions section, the sentence on lines 813-816 ends with a statement that cannot be proven and should be deleted. It is about the following fragment: “that were used to manipulate public opinion”. This should be deleted. Likewise, line 822 states “and their dangerous role in shaping public opinion”. This must also be removed.

The sentence in lines 822-826 should be removed because it is not supported by data. The authors cannot state this if they want to be scientifically rigorous.

Authors should explicitly verify the new hypotheses that they will present in the Conclusions.

In some cases COVID-19 is written in lower case and others in upper case. This must be homogenized.

Author Response

Dear Reviewer,

First of all, I would like to thank you for your assistance in revising and improving the content of the manuscript as well as thorough analysis of each part of it. It is really valuable experience for me. I tried to do my best to tackle the weaknesses and problems that you mentioned and reviewed the paper, paying attention to all your comments, including:

  1. In the Introduction section I added references on the perception of citizens about COVID-19
  2. I revised the research hypotheses, making them more concrete and verifiable with the data from the content and discourse analyses.
  3. I deleted all the statements and references related to shaping public opinion, thank you for this important remark.
  4. I added some comments about the findings in the Results and Discussion section that proved the three research hypotheses.
  5. I changed the order of the Limitations and Conclusions sections according to your review.
  6. I re-wrote the Conclusions section, based on the three hypotheses, making the findings more explicit and verifiable with the data.
  7. I checked all the minor mistakes in English grammar and spelling of 'COVID'.

Thank you very much for your excellent work and assistance!

Reviewer 2 Report

I'm satisfied that this paper clarifies and attenuates in those areas which the previous draft needed. I think this paper deserves publication. 

Author Response

Thank you very much for your assistance in revising my manuscript and attention to each part of it.